# Pertussis Vaccination for Adults: An Updated Guide for Clinicians

**DOI:** 10.3390/vaccines13010060

**Published:** 2025-01-11

**Authors:** Kay Choong See

**Affiliations:** Division of Respiratory and Critical Care Medicine, Department of Medicine, National University Hospital, Singapore 119074, Singapore; kaychoongsee@nus.edu.sg

**Keywords:** *Bordetella pertussis*, diphtheria–tetanus–pertussis vaccine, immunization programs, vaccination hesitancy, whooping cough

## Abstract

Pertussis, or whooping cough, is a highly contagious respiratory infection caused by the Gram-negative bacterium *Bordetella pertussis*. Although traditionally associated with children, pertussis is increasingly prevalent among adults, particularly those with comorbidities or weakened immune systems, where it can lead to severe complications. Diagnosing pertussis in adults can be challenging due to its nonspecific symptoms, underreporting, and the limited sensitivity of available diagnostic tests. While treatment with macrolides is generally effective, it may not significantly alter the clinical course of the disease, and growing concerns about macrolide resistance are emerging. Vaccination remains the cornerstone of prevention, offering proven immunogenicity, efficacy, and safety. However, vaccination uptake remains low, partly due to limited patient awareness and insufficient prioritization by healthcare professionals. This review aims to provide clinicians with critical insights into pertussis epidemiology, vaccination strategies, and the latest guideline recommendations, empowering them to engage in meaningful discussions with adult patients and advocate for increased vaccination to combat this often-overlooked infection.

## 1. Introduction

Pertussis, or whooping cough, is a predominantly acute respiratory tract disease caused by infection with the human-restricted aerobic Gram-negative bacterium *Bordetella pertussis.* It is highly transmissible via respiratory droplets, and an infected individual spreads it to an average of 5.5 other people [1]. The probability that an infection occurs among susceptible people (i.e., secondary attack rate) within household members aged 12 years and over was 11% [2]. After pertussis infection, an individual obtains natural immunity, though this starts to wane after 3.5 years [3].

At least two points along the adult life course have been targeted for pertussis vaccination to decrease the disease burden in infants: during pregnancy (maternal vaccination [4]) and for non-pregnant adults ten or more years after their last pertussis vaccination [5,6]. Maternal vaccination in the second or third trimester stimulates the production of maternal antibodies, which are transferred to the developing fetus and persist into infancy, providing protection against pertussis until infants can be vaccinated at two months of age [4]. Observational studies consistently demonstrate that pertussis vaccination during the second or third trimester of pregnancy is highly effective in providing passive immunity to infants. Maternal pertussis vaccination prevents approximately 70–90% of pertussis cases and up to 90.5% of pertussis-related hospitalizations in infants under three months of age [7]. Additionally, vaccinating non-pregnant adults who are in close contact with unvaccinated young infants helps reduce pertussis transmission, thereby offering indirect protection to these vulnerable infants through a cocooning strategy [8,9].

While pertussis vaccination is crucial for indirectly protecting infants, the importance of vaccinating adults has been relatively overlooked, forming the focus of this review. The uptake of adult pertussis vaccination remains low globally [10]. For instance, in Greece, even though pertussis vaccination is recommended nationally for adults aged 60 years and above, only 1.2% received it when surveyed from March to September 2019 [11]. Even among older adults and patients with pulmonary comorbidity who are at risk of pertussis-related complications, vaccination rates are suboptimal. In an Australian study, among patients aged 65 years and older, only 20.6% were up-to-date with pertussis vaccination [12]. In a U.S. study, among patients aged 65 years and older with asthma and chronic obstructive pulmonary disease, only 25.6% and 19.9%, respectively, received the tetanus, reduced-antigen diphtheria, and reduced-antigen acellular pertussis (Tdap) vaccine [13].

Low uptake is linked to vaccine hesitancy, which may be examined using the World Health Organization (WHO) Strategic Advisory Group of Experts on Immunization’s 3C model [14]: Complacency (overly optimistic perception of personal health status and low perceived vulnerability to infection or severe disease), Confidence (lack of trust in vaccines and the system administering them), and Convenience (structural barriers and accessibility issues). For pertussis, low vaccination uptake is primarily driven by complacency [15], compounded by a lack of awareness about the critical importance of pertussis vaccination, even among healthcare professionals [16,17]. In a survey conducted among physicians and nurses at a teaching hospital, approximately 25% of respondents believed that pertussis affects only children, and just 15% had updated their pertussis vaccination within the past 10 years [16]. Furthermore, even though serious pertussis-related morbidities such as pneumonia, rib fractures, and syncope have been reported among adults [18], health care professionals in another survey did not perceive pertussis as a serious disease among adults, and only 17% felt that adult pertussis vaccination was important [17].

Recognizing the critical role of healthcare provider awareness in driving vaccination behavior and uptake, this review aims to provide clinicians with key insights into pertussis epidemiology, vaccination specifics, and the latest guideline recommendations. By doing so, it seeks to empower clinicians to engage in meaningful discussions with adult patients and advocate for vaccination against this frequently overlooked infection. To incorporate up-to-date information, the PubMed database was searched for articles published between 1 January 2022 and 27 November 2024 using the term “pertussis in adults”. The search was last updated on 2 January 2025. Relevant articles were subsequently added to the author’s personal library for this review.

## 2. Epidemiology of Pertussis in Adults

Pertussis is well recognized among children, though it is less well recognized and underestimated among adults [18]. Given that only *B. pertussis* produces pertussis toxin, the prevalence of disease can be determined using serological surveys that measure pertussis toxin-specific IgG antibodies in the serum. A seroprevalence study conducted among adults aged 40–59 years in 18 European countries revealed that in 13 of these countries, 2.7–5.8% of sera had pertussis toxin antibody levels ≥100 IU/mL, indicative of recent pertussis exposure [19]. These findings suggest the ongoing circulation of pertussis among middle-aged adults, despite the presence of well-implemented childhood vaccination programs. This apparent discordance may be explained using the findings of another survey conducted among adults aged 30 years and older in Greece, which revealed a seroprevalence of up to 21.3% in certain regions [20]. More than 90% of seropositive adults had antibody levels <50 IU/mL, suggesting that most adults are susceptible to further infection.

As an alternative to serological surveys, retrospective studies using administrative databases can be undertaken. In one such study performed among Australian adults aged 50 years and older presenting to primary care, pertussis incidence ranged from 57.6 to 91.4 cases per 100,000 persons between 2015 and 2019 [21]. Incidence rates of pertussis were higher in older adults, in asthmatics, and in patients with chronic obstructive pulmonary disease [21].

A combination of clinical and laboratory methods can be prospectively employed to uncover the prevalence and incidence of disease. An example is a prospective study among patients seen in outpatient clinics and inpatient wards at a Turkish university hospital, where 3.5% of patients with cough duration of between one week and one month had pertussis confirmed with nasopharyngeal swab polymerase chain reaction (PCR) [22]. A similar prevalence of 3.0% was found in primary care adult patients across 12 European countries [23]. Separately, a prospective clinical study of U.S. patients with prolonged cough used enzyme-linked immunosorbent assay (ELISA) to measure IgG antibody levels to pertussis toxin, estimating 202 pertussis cases per 100,000 individuals aged 50–64 years and 257 cases per 100,000 individuals aged 65 years and older [24].

Nonetheless, all epidemiological studies are prone to underestimation for pertussis cases. In adults aged 50 years and over in five Latin American studies, underestimation was calculated as 104–114 times the number of cases reported to national surveillance systems, with the highest number of cases in adults aged 90 years and over [25].

Despite possible under-ascertainment of pertussis cases, increasing numbers are being reported among adults [26,27,28]. According to the European Centre for Disease Prevention and Control (ECDC), 32,037 cases of pertussis were reported in the first quarter of 2024 alone, compared with 25,130 cases for the entirety of 2023 [29]. While most cases and deaths continued to occur in infants, an increased incidence of pertussis was observed across all age groups, including adults. Notably, among the deaths reported to the ECDC in 2023 and early 2024, 42% (8 out of 19) occurred in older adults aged 60 and above. The increasing trend of adult pertussis cases runs counter to the overall decreasing burden of pertussis, particularly childhood infections [30]. Possible reasons for the increasing burden of adult pertussis cases include a lack of lasting immunity after natural infection, waning vaccine-induced immunity post-childhood vaccination, and comparatively low levels of adult pertussis vaccination compared with childhood vaccination.

## 3. Clinical and Economic Impact of Pertussis in Adults

Troublesome pertussis-related symptoms include a prolonged cough, post-tussive vomiting, rhinorrhea, pharyngitis, fever, and fatigue, which may lead to disturbed sleep and decreased quality of life [23,31]. Symptoms may be long-lasting, up to about 160 days for both children and adults [32]. Complications of pertussis include pneumonia, sinusitis, otitis media, cough-induced syncope, subconjunctival hemorrhage, stroke, encephalitis, seizures, urinary incontinence, rectal prolapse, and even mortality [31,33]. In a Canadian study, among adults aged 50 years and older, pertussis caused prolonged cough for 12 weeks or more in 97%, pneumonia in 9%, and urinary incontinence in 34% of women [2].

Adults of advanced age face an increased risk of severe complications and high healthcare resource utilization from pertussis [34,35,36]. Individuals aged 65 years and above are at elevated risk of hospitalization from pertussis-related complications, occurring in up to 14% of adults aged 75 years or older [33]. Death from severe pertussis can also occur in up to 17.4% of adults aged 65 years and over [31]. As of 2019, overall costs are substantial, with an average cost of CAD 1920 per pertussis case in adults aged 65 years and older in Canada [37]. Meanwhile, the total annual economic cost of pertussis was estimated to be approximately GBP 238 million in the United Kingdom [38].

Individuals with asthma, chronic obstructive pulmonary disease, obesity, Crohn’s disease, diabetes mellitus, and immunodeficiency are at an increased risk of severe pertussis resulting in hospitalization [39] and increased economic burden [40]. Between 2009 and 2018 in England, among primary care patients aged 50 years and older with chronic obstructive pulmonary disease, there were 4.73 new cases of pertussis per 100,000 people annually [41]. Pertussis episodes subsequently resulted in approximately 2.4 times more general practitioner/nurse visits, 10 times more emergency department visits, and an increase in medical costs of over GBP 4000 per patient [41].

The clinical impact of pertussis extends beyond the acute infection. In the U.S., between 2007 and 2019, pertussis infections were associated with an increased risk of asthma and chronic obstructive pulmonary disease exacerbations, spanning from 30 days prior to diagnosis to 180 days afterward [42].

## 4. Diagnosing and Treating Pertussis in Adults

Four clinical features are useful in the diagnosis of pertussis in adults: paroxysmal cough, post-tussive vomiting, inspiratory whooping, and absence of fever. To help rule out pertussis, look for the absence of paroxysmal cough and presence of fever, given the following statistics for paroxysmal cough (sensitivity 93.2%, specificity 20.6%) and absence of fever (sensitivity 81.8%, specificity 18.8%). To help rule in pertussis, look for inspiratory whooping (sensitivity 32.5%, specificity 77.7%) and post-tussive vomiting (sensitivity 29.8%, specificity 79.5%) [43].

Confirming *B. pertussis* infection is usually carried out by obtaining nasopharyngeal aspirates or swabs and performing PCR. Although this test has high sensitivity (>90%) and specificity (>90%) early in disease, and has a rapid turnaround time, sensitivity falls in the later stages of infection [44,45]. Other tests are less relevant for clinical practice. Pertussis cultures have slow turnaround times and have low sensitivity (64.0%) but can act as reference standards for research [45]. Serological assays for antibodies against pertussis toxin can be performed on serum, enabling the distinction between recent and past infections [45]. While these assays may be valuable for epidemiological surveys, they have notable limitations, since they require time for serial testing and cannot distinguish between antibodies generated by natural infection and those induced by vaccination.

Delayed diagnosis of pertussis in adults is common, with one study utilizing U.S. insurance claim data reporting an average delay of 13.8 days [46]. Delayed diagnosis of pertussis in adults is often due to several diagnostic challenges. The disease frequently presents with nonspecific symptoms and atypical features in adults, making it difficult to distinguish from other respiratory illnesses [43]. Many cases go untested, and even when testing is performed, the sensitivity of available methods is limited, especially in adults presenting late in the disease course [44,45]. Additionally, there is low awareness of pertussis as a potential diagnosis among adults, contributing to underreporting, particularly in settings where testing is not routinely conducted [18].

For patients with pertussis, short-term antibiotics such as azithromycin (3–5 days), clarithromycin or erythromycin (7 days), or trimethoprim/sulfamethoxazole (7 days) effectively eradicate *B. pertussis* from the nasopharynx. However, they may not significantly alter the clinical course of the illness [47]. Additionally, while *B. pertussis* remains largely susceptible to macrolides, emerging resistance has been reported in China [48] and France [49].

## 5. Efficacy and Safety of Vaccination in Adults

Given the diagnostic and therapeutic challenges of pertussis, vaccination remains the cornerstone strategy for reducing its disease burden. Pertussis vaccination in adults is essentially booster vaccination after the primary series undertaken during infancy. The immune response to *B. pertussis* can be assessed by measuring antibody concentrations against various pertussis antigens: pertussis toxin (also conducted for seroprevalence surveys), filamentous hemagglutinin, fimbriae type 2, fimbriae type 3, and pertactin. However, a definitive correlation with protection has not been established and therefore the protection afforded by pertussis vaccination is best assessed by the proportion of infections avoided in randomized trials (i.e., vaccine efficacy), and in real-world observational studies (i.e., vaccine effectiveness).

The introduction of the combined diphtheria–tetanus–whole-cell pertussis (DTwP) vaccination in the 1950s significantly reduced the global burden of pertussis, but safety concerns from high reactogenicity (i.e., the capacity of a vaccine to produce adverse reactions) prompted a switch to acellular pertussis vaccines in the 1990s [50]. However, this comes at the expense of reduced vaccine efficacy and a shorter duration of disease protection with acellular pertussis vaccines compared with whole-cell vaccines [51,52]. Infant formulations involve full doses of diphtheria and pertussis antigens (DTaP), while adult formulations employ a reduced content of these two antigens (Tdap) to avoid increasing reactogenicity with successive vaccinations [53]. Two adult formulations of Tdap have been developed for use in adolescents and adults: Adacel^®^ (Sanofi Pasteur, Lyon, France) and Boostrix^®^ (GlaxoSmithKline, London, UK) (Table 1).

The immunogenicity of the Tdap vaccine is satisfactory. A single 0.5 mL intramuscular dose of Adacel^®^ produced pertussis antibody levels 2.1 to 5.4 times higher than those generated by the three-dose primary series administered at 2, 4, and 6 months of age [54]. A single 0.5 mL intramuscular dose of Boostrix^®^ generated antibody responses against pertussis antigens in >90% of individuals, with geometric mean concentrations increasing 5- to 44-fold a month post-vaccination [53]. Across 17 clinical trials, pertussis booster vaccination provided good immunogenicity (>85% antibody response), good vaccine efficacy (reduced pertussis cases by 92% in individuals with versus without booster vaccination over two years, according to one trial [55]), and acceptable safety (side effects were few and minor such as rare nausea and vomiting) [26].

Current acellular pertussis vaccines may be less effective in protecting individuals against pertactin-deficient *B. pertussis* compared with whole-cell pertussis vaccines. Acellular pertussis vaccines also do not prevent the colonization or transmission of *B. pertussis* unlike prior natural infection [50]. Despite these drawbacks, real-world vaccine effectiveness with acellular pertussis vaccines was shown to be moderately high at 53–64% in a case-control study performed in northern California [56].

Pertussis vaccination is generally well tolerated. In a systematic review of pertussis vaccination in adults aged 50 years and above, the more common adverse events are minor ones, including injection site pain (reported in 21.5–78% of vaccinated individuals), fatigue (13–23.2%), headache (42.6%), and myalgia (19.4–23%); serious adverse events such as anaphylaxis and neuroinflammatory disorders are rare (<1%) [18]. On a similar note, a descriptive study conducted in São Paulo, Brazil, from 2015 to 2016 evaluated adverse events following immunization (AEFI) with the Tdap vaccine during pregnancy and its results supported the safety of Tdap among pregnant mothers [57]. A total of 48 of 201 (23.9%) mothers reported at least one AEFI, with 60 symptoms identified. The most common were pain (22.4%), swelling (2.5%), fever (1.5%), and somnolence (1.0%) with other minor symptoms each occurring in less than 1% of cases. Reassuringly, a systematic review of routine immunization in the United States found no evidence of an increased risk of stillbirth associated with maternal Tdap vaccination [58].

**Table 1 vaccines-13-00060-t001:** Pertussis vaccines for adults.

Vaccine	Manufacturer	Vaccine Type	Administration	Author (Year) [ref]
Adecel^®^	Sanofi Pasteur (Lyon, France)	Aluminum phosphate-adjuvanted tetanus, reduced-antigen diphtheria, 5-component (pertussis toxin, filamentous hemagglutinin, fimbriae type 2, fimbriae type 3, pertactin) reduced-antigen acellular pertussis combination vaccine (Tdap)	Intramuscular, 0.5 mL per dose	Pichichero (2005) [54]
Boostrix^®^	Glaxo-SmithKline (London, UK)	Aluminum hydroxide-adjuvanted tetanus, reduced-antigen diphtheria, 3-component (pertussis toxin, filamentous hemagglutinin, pertactin) reduced-antigen acellular pertussis combination vaccine (Tdap). Note that the aluminum adjuvant content may differ in various countries [53], without affecting immunogenicity [59]	Intramuscular, 0.5 mL per dose	Ward (2005) [55] (also known as the APERT Study)

APERT: Adult Pertussis Trial. Tdap: tetanus, reduced-antigen diphtheria, and reduced-antigen acellular pertussis.

## 6. Vaccine Co-Administration and Re-Vaccination in Adults

Co-administration of Tdap with other vaccines, including those for hepatitis A, influenza, human papillomavirus, meningococcus, pneumococcus, and poliovirus, has been shown to be both safe and immunogenic [53]. In a study of 772 Chinese applicants for a United States immigrant visa, who simultaneously received the *Haemophilus influenzae* type b conjugate vaccine, oral polio vaccine, hepatitis B vaccine, combined measles–mumps–rubella vaccine, varicella vaccine, influenza vaccine, pneumococcal polysaccharide vaccine, and combined diphtheria, tetanus toxoid, and acellular pertussis vaccine, 49.6% of individuals reported adverse reactions within seven days post-vaccination [60]. These adverse reactions were mild (mainly injection site pain and fever) and resolved within 72 h.

Co-administration of Tdap with the newer COVID-19 and respiratory syncytial virus vaccines appears to pose no significant issues. In a case report involving co-administration of the BNT162b2 mRNA COVID-19 vaccine in a 29-year-old Asian woman, the development of SARS-CoV-2 spike antibodies was delayed till 8 weeks after the second dose, but eventual immune response remained adequate [61]. In a randomized trial examining the co-administration of a prefusion F subunit respiratory syncytial virus vaccine and Tdap, while the immunogenicity of the pertussis component of Tdap was reduced (the reason for which remains unknown), it was still considered protective [62].

Regarding the co-administration of the pertussis vaccine with other vaccines, it may seem logical to expect greater reactogenicity and side effects compared with separate administration. However, this is not always the case. An observational study conducted between 2020 and 2022 in Hebei Province, China, found that adverse events following immunization were significantly lower—66.9 per million doses—when the diphtheria, tetanus toxoid, acellular pertussis, inactivated poliovirus, and *Haemophilus influenzae* type b vaccines were administered as a combination vaccine [63]. In contrast, the rate was higher at 637.8 per million doses when these vaccines were administered separately. This difference was attributed to the combination vaccine’s ability to reduce the number of injections and the amount of vaccine adjuvant required.

The re-vaccination interval of Tdap may be performed once every 10 years (i.e., decennial vaccination) [5,6]. In a serological survey of 6060 healthy children across five Chinese provinces, investigators found that pertussis antibodies begin to wane shortly after vaccination with a primary series at 4–6 months and a booster at 18–24 months of age, with natural infection rates increasing from the age of six [64]. In contrast, follow-up of antibody persistence among adult participants of a randomized clinical trial suggest that re-vaccination can be administered every 10 years [54,65]. Even though vaccine-induced pertussis-toxin antibodies drop to pre-vaccination levels in adults five years post-vaccination, vaccine-induced antibodies against filamentous hemagglutinin, fimbriae type 2, fimbriae type 3, and pertactin persisted for at least 10 years [65]. Repeat Tdap vaccination with either Adacel^®^ or Boostrix^®^ 10 years after the first booster vaccination was immunogenic and well tolerated [66,67].

## 7. Management of Side Effects of Pertussis Vaccination

After pertussis vaccination, the most common adverse events following immunization (AEFI) include injection site pain and swelling, myalgia, fever, fatigue, somnolence, and headache [18,57]. Severe AEFI including syncope, seizures, and anaphylaxis can rarely occur [68,69,70].

The management of these AEFI is supportive and follows general treatment principles, as no pertussis vaccine-specific interventions have been studied. Although syncope rarely results in serious traumatic injury, it is reasonable to monitor vaccinated patients for at least 15 min, with additional observation until symptoms resolve if syncope occurs [4].

## 8. Guideline Recommendations for Pertussis Vaccination in Adults

The WHO does not provide definitive recommendations for pertussis vaccination, and national guidelines for adult pertussis vaccination vary by country [71]. For instance, according to the Greek National Adult Vaccination Program, adults aged 19 to 25 should receive a single booster dose of the Tdap vaccine, followed by booster doses of Tdap every 10 years [20]. However, less than half of Europe has vaccination policies for pertussis [71]. Like the variability in national guidelines, recommendations from major professional bodies either support pertussis vaccination or do not address it (Table 2).

## 9. Improving the Uptake of Adult Pertussis Vaccination

To enhance adult pertussis vaccination uptake, insights from vaccine hesitancy observed during the COVID-19 pandemic, particularly after the end of the public health emergency of international concern, can guide effective strategies. At the physician level, these include the following: thoughtful message framing, persuasive communication emphasizing safety and personal or societal benefits, sharing personal stories, creating safe spaces for open dialogue, leveraging co-administration with annual influenza vaccines, and employing decision aids and visual tools to support informed decision making [80]. In other words, healthcare professionals need to boost awareness, actively recommend immunization especially among at-risk patients, and improve the ease of obtaining the pertussis vaccine [10].

As mentioned in the Introduction, the low uptake of vaccination in general is linked to vaccine hesitancy, which may be examined using the World Health Organization (WHO) Strategic Advisory Group of Experts on Immunization’s 3C model (Complacency, Confidence, Convenience) [14]. Using this framework, at the individual level, clinicians should consider recommending pertussis vaccination for older adults and individuals with comorbidities, emphasizing shared decision making to address key considerations and ensure informed choices. Table 3 summarizes the key considerations and discussion topics that clinicians should cover when counseling adults for pertussis vaccination.

Beyond the individual level, system-level strategies to improve the uptake of pertussis vaccination can be explored. These strategies include targeted messaging, mass media health communication campaigns, on-site vaccine availability, pharmacist-led administration, integration into healthcare protocols, incentive programs, and the use of chatbots to enhance accessibility and engagement [80]. Interventions that integrate both individual-level and system-level strategies are likely to be more effective in improving adult vaccination uptake than single approaches [81].

Local infectious disease surveillance systems need to be broadened to understand the true burden of pertussis, while rates of pertussis vaccination need to be tracked regionally. Updated insights into the burden of pertussis and the latest data on pertussis vaccination should be regularly communicated to all clinicians, including primary care physicians, specialist clinicians, and clinicians-in-training. Effective communication can be facilitated through various teaching platforms, including continuing medical education workshops, webinars, and asynchronous methods such as pre-recorded lectures, self-administered quizzes, and relevant reading materials in medical newsletters. These medical education efforts aim not only to enhance clinicians’ awareness and knowledge of the importance of effective patient counseling but also to improve clinical practice performance and ultimately achieve better clinical outcomes [82,83,84]. Ideally, these initiatives should be complemented by a medical school curriculum that incorporates immunization teaching to strengthen preventive medicine skills from the outset [85].

Once the healthcare and economic burden of pertussis is fully understood, national immunization guidelines can be strengthened to definitively recommend vaccination and secure funding. To strike an optimal balance between societal benefits and vaccination costs, adopting a focused and targeted approach may prove to be a pragmatic solution. For example, strongly recommending a single dose of Tdap at the age of 65 years, administered alongside other age-appropriate vaccines, could be prioritized over attempting to implement decennial booster vaccinations for the entire population [86].

## 10. Future Directions

Pertussis vaccination may also be further enhanced to prevent both severe disease and bacterial colonization, with the latter not well addressed by current Tdap vaccines [50]. Nasal pertussis vaccines, such as BPZE1, which contains a genetically inactivated strain of *B. pertussis*, are under development to offer dual benefits: protection against the disease and prevention of nasal colonization by inducing *B. pertussis*-specific mucosal secretory IgA responses [87,88]. The latter addresses a significant limitation of current acellular pertussis vaccines, which do not generate significant IgA responses [50].

Another concern is that replacing the whole-cell pertussis vaccine with the acellular pertussis vaccine could lead to an increased mutation rate of *B. pertussis* [89], potentially giving rise to resistant strains, such as those deficient in pertactin [50,90]. The reason for this is not entirely certain and has been ascribed to the acellular vaccine being less effective than the whole-cell vaccine, with a shorter duration of protection and an inability to prevent colonization or transmission of *B. pertussis* [50]. Nonetheless, this potential evolutionary change could contribute to outbreaks of resistant pertussis, even in populations with high vaccination coverage. As a result, improved pertussis vaccine formulations, adjuvants, and schedules are needed to optimize both effectiveness and reactogenicity [91].

In addition, multi-pathogen combination or conjugate vaccines could be developed to include a pertussis component, enhancing convenience, acceptability, and potentially reducing the occurrence of side effects [63]. Other conceivable advantages of combination vaccines include logistical savings (fewer syringes; reduced packaging; decreased disposal needs; less cold chain storage and transportation space required) and a more efficient use of human resources (shorter administration times with fewer errors; lower risk of needlestick injuries) [92].

Finally, studies could be conducted to assess the public health benefits of adult vaccination. These benefits may extend beyond protecting older adults from severe pertussis, possibly contributing to herd immunity that helps shield younger children from pertussis infection [93].

## 11. Conclusions

In conclusion, pertussis is often underrecognized, significantly contributing to the disease burden, particularly among older adults and those with comorbid conditions. Effective prevention is achievable through vaccines, which demonstrate high efficacy, strong real-world effectiveness, and minimal adverse effects. However, both naturally acquired and vaccine-induced immunity wane over time, necessitating decennial vaccination. Furthermore, adult pertussis vaccination can protect others by contributing to herd immunity and through a cocooning approach, where adults in close contact with unvaccinated infants under two months old help reduce the risk of transmitting pertussis to these vulnerable infants. Therefore, to reduce the burden of pertussis, it is crucial for clinicians to actively assess the risk of severe disease, engage in shared decision making, and ensure easy access to vaccination for these adult patients.

## Figures and Tables

**Table 2 vaccines-13-00060-t002:** Selected guideline recommendations for pertussis vaccination in adults.

Patient Population	Guideline (Year) [ref]	Recommendations for Pertussis Vaccination in Adults
All adults	World Health Organization (WHO) (2015) [72]	No definite recommendation Decision to vaccinate with Tdap depends on local epidemiology and healthcare burden of pertussis
All adults	U.S. Centers for Disease Control and Prevention (CDC) (2024) [5,6]	For adults aged 19 years and older, who had received childhood pertussis vaccination, administer one dose of Tdap, followed by Tdap booster every 10 years For pertussis vaccine-naïve adults, administer one dose of Tdap followed by one dose of Tdap at least 4 weeks later, and a third dose of Tdap 6–12 months later, then Tdap booster every 10 years
Adults with asthma	Global Initiative for Asthma (GINA) (2024) [73]	Adults and the elderly with asthma are encouraged to follow local immunization schedules for pertussis vaccination
Adults with cancer	American Society of Clinical Oncology (ASCO) (2024) [74]	For adults aged 19 years and older, who had received childhood pertussis vaccination, administer one dose of Tdap, followed by Tdap booster every 10 years For pertussis vaccine-naïve adults, administer one dose of Tdap followed by one dose of Tdap at least 4 weeks later, and a third dose of Tdap 6–12 months later, then Tdap booster every 10 years For adult hematopoietic stem cell transplant recipients, patients lose immunity from childhood vaccinations and require the vaccination regime for pertussis vaccine-naïve adults, starting 6–12 months after transplant regardless of ongoing immunosuppression
Adults with chronic coronary disease	Multi-society * (2023) [75], European Society of Cardiology (ESC) (2024) [76]	While influenza, COVID-19, and pneumococcal vaccinations are recommended, pertussis vaccination is not mentioned in the multi-society * guidelines The ESC guidelines recommend “vaccination against influenza, pneumococcal disease and other widespread infections, e.g., COVID-19”, but make no specific mention of pertussis vaccination
Adults with chronic kidney disease	Kidney Disease: Improving Global Outcomes (KDIGO) (2024) [77]	Recommends a vaccination program, though no specific mention for any vaccine
Adults with chronic obstructive pulmonary disease	Global Obstructive Lung Disease (GOLD) (2025) [78]	Follows CDC recommendations for chronic obstructive pulmonary disease patients to receive Tdap vaccination
Adults with diabetes mellitus	American Diabetes Association (ADA) (2024) [79]	Follows CDC recommendations for diabetic patients to receive Tdap vaccination

Tdap: tetanus, reduced-antigen diphtheria, and reduced-antigen acellular pertussis. * American Heart Association, American College of Cardiology, American College of Clinical Pharmacy, American Society for Preventive Cardiology, National Lipid Association, and Preventive Cardiovascular Nurses Association.

**Table 3 vaccines-13-00060-t003:** Key topics for clinical conversations on adult pertussis vaccination.

Domain *	Key Topics for Clinical Conversations
Complacency	Highlight the rising incidence of pertussis in adults, complications such as troublesome cough and pneumonia, and increased cost of care Highlight waning immunity after both naturally acquired infections and prior vaccination, with little protection 10 years post-vaccination
Confidence	Assure that pertussis vaccination with Tdap is effective, with real-world protection levels exceeding 50% of cases avoided Reassure that side effects usually involve transient injection site pain and fever that resolve within 72 h
Convenience	Allow for co-administration with most other vaccines including influenza, pneumococcal, COVID-19, and respiratory syncytial virus vaccines

Tdap: tetanus, reduced-antigen diphtheria, and reduced-antigen acellular pertussis. * Following the World Health Organization (WHO) Strategic Advisory Group of Experts on Immunization’s 3C model [14].

## Data Availability

All data used can be found in the text and tables.

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
