# Peer review of "Pertussis Vaccination for Adults: An Updated Guide for Clinicians"

_vaccines, 2025, doi:10.3390/vaccines13010060_

Round 1

Reviewer 1 Report

Comments and Suggestions for Authors

Dear authors

Thank you for the opportunity to review the Literature review manuscript entitled “Pertussis Vaccination for Adults. An Updated Guide for Clinicians”.

My decision is Accept after major revision.

The paper is interesting and deal with the emerging topic especially for the European Region after the recent pertussis outbreak. I think the major issue is the methodology of the manuscript. Author should be describing their selection criteria for studies to be involved in this review.

Abstract

Add the methodology and sources of your manuscript and the period of searching.

Introduction

According to ECDC during 2023-24, in 17 EU/EEA countries, infants (those under the age of one year) represented the group with the highest reported incidence, whereas in six countries, the highest incidence is reported in adolescents 10-19 years. The majority of deaths occurred in infants. However, increase cases of pertussis were observed to adult populations also. Insert to section data about the recent epidemic crisis of pertussis in Europe.

Discussion

·          The author should be discussed and other strategies for adults to eliminate pertussis such a “cocooning strategy” or “maternal immunization”: Suggested references

·          MMWR Morb Mortal Wkly Rep. 2011;60(41):1424–1426.

·          Prevention of pertussis, tetanus, and diphtheria among pregnant and postpartum women and their infants: Recommendations of the Advisory Committee on Immunization Practices (ACIP). MMWR Recomm Rep 57:1–51, May 30, 2008. Erratum in MMWR Morb Mortal Wkly Rep 57:723, Jul. 4, 2008. http://www.cdc.gov/mmwr/PDF/rr/rr5704.pdf

·          Forsyth K.D., et al.: Prevention of pertussis: Recommendations derived from the second Global Pertussis Initiative roundtable meeting. Vaccine 25:2634–2642, Mar. 30, 2007.

·          Please discussed about the optimal timing of Tdap vaccination during pregnancy to maximize   protection mother and the newborn. Suggested references:             https://doi.org/10.1080/14760584.2020.1791092

·          Add and discussed the low vaccination coverage of adults  and exam the possibility be a potential factor for the recent pertussis epidemic in Europe: reference https://doi.org/10.1038/s41467-021-23114-y    and  reference [20].

·          According to title I suggest to author to enhance the manuscript with a section to be refer how could a clinician to confront the basic side effects of pertussis vaccinations: Suggested reference. http://www.cdc.gov/mmwr/PDF/rr/rr5704.pdf .

Section 5. Efficacy and safety of vaccination in adults

Please add information’s and discus the path of Whole-Cell Vaccines to acecullar vaccines. Indicate the main reasons and the benefits and limitations of the using of acecullar vaccines. Tdap and TDap vaccine help to protect for the same disease but are used for a different group. Indicate the differs and reasons those given to adults in that it is administered as a single dose and has a higher concentration of pertussis antigens.

Conclusions

Enhance the section with more specific conclusions about the necessity of Pertussis Vaccination for Adults.

References

Please reform according to Journal suggestions

Author Response

Dear authors

Thank you for the opportunity to review the Literature review manuscript entitled “Pertussis Vaccination for Adults. An Updated Guide for Clinicians”.

My decision is Accept after major revision.

The paper is interesting and deal with the emerging topic especially for the European Region after the recent pertussis outbreak. I think the major issue is the methodology of the manuscript. Author should be describing their selection criteria for studies to be involved in this review.

Reply: Since this is a narrative review, a systematic search and selection process was not conducted. However, I reviewed the literature to incorporate relevant updates. To incorporate up-to-date information, the PubMed database was searched for articles published between 2022 and 2024 using the term “pertussis in adults.” The initial search was conducted on 27 November 2024, and updated on 2 January 2025, in conjunction with the manuscript revision. Relevant articles were subsequently added to the author’s personal library for this review.

Abstract

Add the methodology and sources of your manuscript and the period of searching.

Reply: This has been added to the Introduction.

Introduction

According to ECDC during 2023-24, in 17 EU/EEA countries, infants (those under the age of one year) represented the group with the highest reported incidence, whereas in six countries, the highest incidence is reported in adolescents 10-19 years. The majority of deaths occurred in infants. However, increase cases of pertussis were observed to adult populations also. Insert to section data about the recent epidemic crisis of pertussis in Europe.

Reply: Included new data from Europe in the section “Epidemiology of pertussis in adults”

Discussion

  • The author should be discussed and other strategies for adults to eliminate pertussis such a “cocooning strategy” or “maternal immunization”: Suggested references
  • MMWR Morb Mortal Wkly Rep. 2011;60(41):1424–1426.
  • Prevention of pertussis, tetanus, and diphtheria among pregnant and postpartum women and their infants: Recommendations of the Advisory Committee on Immunization Practices (ACIP). MMWR Recomm Rep 57:1–51, May 30, 2008. Erratum in MMWR Morb Mortal Wkly Rep 57:723, Jul. 4, 2008. http://www.cdc.gov/mmwr/PDF/rr/rr5704.pdf
  • Forsyth K.D., et al.: Prevention of pertussis: Recommendations derived from the second Global Pertussis Initiative roundtable meeting. Vaccine 25:2634–2642, Mar. 30, 2007.

Reply: Thank you for the suggestions. These have been added.

  • Please discussed about the optimal timing of Tdap vaccination during pregnancy to maximize protection mother and the newborn. Suggested references:             https://doi.org/10.1080/14760584.2020.1791092

Reply: Thank you for the suggestion. This has been added.

  • Add and discussed the low vaccination coverage of adults and exam the possibility be a potential factor for the recent pertussis epidemic in Europe: reference https://doi.org/10.1038/s41467-021-23114-y    and  reference [20].

Reply: Thank you for the suggestion. This has been added.

  • According to title I suggest to author to enhance the manuscript with a section to be refer how could a clinician to confront the basic side effects of pertussis vaccinations: Suggested reference. http://www.cdc.gov/mmwr/PDF/rr/rr5704.pdf .

Reply: Thank you for the suggestion. This has been added.

Section 5. Efficacy and safety of vaccination in adults

Please add information’s and discus the path of Whole-Cell Vaccines to acecullar vaccines. Indicate the main reasons and the benefits and limitations of the using of acecullar vaccines. Tdap and TDap vaccine help to protect for the same disease but are used for a different group. Indicate the differs and reasons those given to adults in that it is administered as a single dose and has a higher concentration of pertussis antigens.

Reply: Thank you for the suggestion. These points have been clarified.

Conclusions

Enhance the section with more specific conclusions about the necessity of Pertussis Vaccination for Adults.

Reply: Thank you for the suggestion. These points have been clarified.

References

Please reform according to Journal suggestions

Reply: I have not yet received any suggestions from the Journal but will make any necessary revisions upon receiving feedback from the editorial staff.

Reviewer 2 Report

Comments and Suggestions for Authors

This was an excellent review on Pertussis vaccinations. The manuscript was well-written and easy to read. I didn't find any issues that would require revisions. The manuscript can be accepted for publication in Vaccines. 

Author Response

This was an excellent review on Pertussis vaccinations. The manuscript was well-written and easy to read. I didn't find any issues that would require revisions. The manuscript can be accepted for publication in Vaccines.

Reply: Thank you for your encouraging comments.

Reviewer 3 Report

Comments and Suggestions for Authors

Thank you for sharing your guide for clinicians on pertussis vaccination for adults. The following minor comments may help when revising the manuscript.

L32-34: Please state in your manuscript the year/time period during which 1.2% adults aged 60 years and above received the vaccine.

L35-36: What did the patients suffer from that made them being at higher risk; please revise for more clarity.

General comment: Regarding the flow of your article, please consider presenting the section "epidemiology of pertussis in adults" before the section "diagnosis  and treating pertussis in adults". 

Table 3: You raised the issues of complacency, confidence and convenience first time in the instruction. It is not quite clear why you present Table 3 at the end of your manuscript. 

Author Response

Thank you for sharing your guide for clinicians on pertussis vaccination for adults. The following minor comments may help when revising the manuscript.

Reply: Thank you for your kind comments.

L32-34: Please state in your manuscript the year/time period during which 1.2% adults aged 60 years and above received the vaccine.

Reply: Stated that the survey was conducted from March to September 2019.

L35-36: What did the patients suffer from that made them being at higher risk; please revise for more clarity.

Reply: Clarified as older adults and patients with pulmonary comorbidity.

General comment: Regarding the flow of your article, please consider presenting the section "epidemiology of pertussis in adults" before the section "diagnosis  and treating pertussis in adults".

Reply: The sections are re-arranged as suggested.

Table 3: You raised the issues of complacency, confidence and convenience first time in the instruction. It is not quite clear why you present Table 3 at the end of your manuscript.

Reply: Clarified that Table 3 summarizes the key considerations and discussion points for clinicians when counselling adults about pertussis vaccination. This table, along with the accompanying text, has been included in a new section titled “Improving the uptake of adult pertussis vaccination.”

Reviewer 4 Report

Comments and Suggestions for Authors

As this review article is on pertussis vaccine, it should focus more on the vaccine, rather than the diseases.  Section 2-4 can be shortened and combined.

I will also suggest to add a separate talking about the low acceptance rate of the vaccine, the possible  solutions etc. 

The review article should be much more focused, and should not talk too much on the background of the infection.

Please consider revise it to focus on vaccination, to concur with the title as well as the journal.

Author Response

As this review article is on pertussis vaccine, it should focus more on the vaccine, rather than the diseases.  Section 2-4 can be shortened and combined.

Reply: In response to the Editorial Office's request for a longer manuscript with additional details and recognizing the importance of understanding pertussis for effective patient counselling on vaccination, I did not further shorten the manuscript. Instead, I expanded it as suggested, incorporating more comprehensive information about the vaccine based on the valuable feedback from all reviewers.

I will also suggest to add a separate talking about the low acceptance rate of the vaccine, the possible  solutions etc.

Reply: Added a new section titled “Improving the uptake of adult pertussis vaccination.”

The review article should be much more focused, and should not talk too much on the background of the infection. Please consider revise it to focus on vaccination, to concur with the title as well as the journal.

Reply: In response to the Editorial Office's request for a longer manuscript with additional details and recognizing the importance of understanding pertussis for effective patient counselling on vaccination, I did not further shorten the manuscript. Instead, I expanded it as suggested, incorporating more comprehensive information about the vaccine based on the valuable feedback from all reviewers.

Round 2

Reviewer 1 Report

Comments and Suggestions for Authors

Dear author thank you for your revisions .My decision is accept.